# Closed-Loop Control of Additive Manufacturing via Reinforcement Learning

## Abstract

Additive manufacturing suffers from imperfections in hardware control and material consistency. As a result, the deposition of a wide range of materials requires on-the-fly adjustment of process parameters. Unfortunately, learning the in-process control is challenging. The deposition parameters are complex and highly coupled, artifacts occur after long time horizons, available simulators lack predictive power, and learning on hardware is intractable. In this work, we demonstrate the feasibility of learning a closed-loop control policy for additive manufacturing. To achieve this goal, we assume that the perception of a deposition device is limited and can capture the process only qualitatively. We leverage this assumption to formulate an efficient numerical model that explicitly includes printing imperfections. We further show that in combination with reinforcement learning, our model can be used to discover control policies that outperform state-of-the-art controllers. Furthermore, the recovered policies have a minimal sim-to-real gap. We showcase this by implementing a single-layer self-correcting printer.

## 1 Introduction

A critical component of manufacturing is identifying process parameters that consistently produce, high-quality structures. In commercial devices, this is typically achieved by expensive trial-and-error experimentation (Gao et al., 2015). To make such an optimization feasible, a critical assumption is made: there exists a set of parameters for which the relationship between process parameters and process outcome is predictable. However, such an assumption does not hold in practice because all manufacturing processes are stochastic in nature. Specifically in additive manufacturing, variability in both materials and intrinsic process parameters can cause geometric errors leading to imprecision that can compromise the functional properties of the final prints. Therefore, transition to closed-loop control is indispensable for industrial adoption of additive manufacturing (Wang et al., 2020).

Recently, we have seen promising progress in learning policies for interaction with amorphous materials (Li et al., 2019b; Zhang et al., 2020). Unfortunately, in the context of additive manufacturing, discovering effective control strategies is significantly more challenging. The deposition parameters have a non-linear coupling to the dynamic material properties. To assess the severity of deposition errors, we need to observe the material over long time horizons. Available simulators either lack predictive power (Mozaffar et al., 2018) or are too complex for learning (Tang et al., 2018; Yan et al., 2018). Moreover, learning on hardware is intractable as we require tens of thousands of printed samples. These challenges are further exaggerated by the limited perception of printing hardware, where typically, only a small in-situ view is available to assess the deposition quality.

In this work, we propose the first closed-loop controller for additive manufacturing based on reinforcement learning deployed on real hardware. To achieve this we formulate a custom numerical model of the deposition process. Motivated by the limited hardware perception we make a key assumption: to learn closed-loop control it is sufficient to model the deposition only qualitatively. This allows us to replace physically accurate but prohibitively slow simulations with efficient approximations. To ameliorate the sim-to-real gap, we enhance the simulation with a data-driven noise distribution on the spread of the deposited material. We further show that careful selection of input and action space is necessary for hardware transfer. Lastly, we leverage the privileged information about the deposition process to formulate a reward function that encourages policies that account for material changes over long horizons. Thanks to the above advancements, our control policy can

be trained exclusively in simulation with a minimal sim-to-real gap. We demonstrate that our policy outperforms baseline deposition methods in simulation and physical hardware with low or high viscosity materials. Furthermore, our numerical model can serve as an essential building block for future research in optimal material deposition, and we plan to make the source code available.

## 2 RELATED WORK

To identify process parameters for additive manufacturing, it is important to understand the complex interaction between a material and a deposition process. This is typically done through trial-and-error experimentation (Kappes et al., 2018; Wang et al., 2018; Baturynska et al., 2018). Recently, optimal experiment design and, more specifically, Gaussian processes have become a tool for efficient use of the samples to understand the deposition problem (Erps et al., 2021). However, even though Gaussian Processes model the deposition variance, they do not offer tools to adjust the deposition on-the-fly. Another approach to improve the printing process is to design closed-loop controllers. One of the first designs was proposed by Sitthi-Amorn et al. (2015) that monitors each layer deposited by a printing process to compute an adjustment layer. Liu et al. (2017) built upon the idea and trained a discriminator that can identify the type and magnitude of observed defects. A similar approach was proposed by Yao et al. (2018) that uses handcrafted features to identify when a print significantly drops in quality. The main disadvantage of these methods is that they rely on collecting the in-situ observations to propose one corrective step by adjusting the process parameters. However, this means that the prints continue with sub-optimal parameters, and it can take several layers to adjust the deposition. In contrast, our system runs in-process and reacts to the in-situ views immediately. This ensures high-quality deposition and adaptability to material changes.

Recently machine learning techniques sparked a new interest in the design of adaptive control policies (Mnih et al., 2015). A particularly successful approach for high-quality in-process control is to adopt the Model Predictive Control paradigm (MPC) (Gu et al., 2016; Silver et al., 2017; Oh et al., 2017; Srinivas et al., 2018; Nagabandi et al., 2018). The control scheme of MPC relies on an observation of the current state and a short-horizon prediction of the future states. By manipulating the process parameters, we observe the changes in future predictions and can pick a future with desirable characteristics. Particularly useful is to utilize deep models to generate differentiable predictors that provide derivatives with respect to control changes (de Avila Belbute-Peres et al., 2018; Schenck & Fox, 2018; Toussaint et al., 2018; Li et al., 2019a). However, addressing the uncertainties of the deposition process with MPC is challenging. In a noisy environment, we can rely only on the expected prediction of the deposition. This leads to a conservative control policy that effectively executes the mean action. Moreover, reacting to material changes over time requires optimizing actions for long time horizons which is a known weakness of the MPC paradigm (Garcia et al., 1989). As a result, MPC is not suitable for in-process control in noisy environments.

Another option to derive control policies is to leverage deep reinforced learning (Rajeswaran et al., 2017; Liu & Hodgins, 2018; Peng et al., 2018; Yu et al., 2019; Lee et al., 2019; Akkaya et al., 2019). The key challenge in the design of such controllers is formulating an efficient numerical model that captures the governing physical phenomena. As a consequence, it is most commonly applied to rigid body dynamics and rigid robots where such models are readily available (Todorov et al., 2012; Bender et al., 2014; Coumans & Bai, 2016; Lee et al., 2018). In contrast, learning with non-rigid objects is significantly more challenging as the computation time for deformable materials is higher and relies on some prior knowledge on the task (Clegg et al., 2018; Elliott & Cakmak, 2018; Ma et al., 2018; Wu et al., 2019). Recently Zhang et al. (2020) proposed a numerical model for training control policies where a rigid object interacts with amorphous materials. Similarly, in our work a rigid printing nozzle interacts with the fluid-like printing material. However, our model is specialized for the printing hardware and models not only the deposition but also its variance. We demonstrate that this is an important component in minimizing the sim-to-real gap and design control policies that are readily applicable to the physical hardware.

## 3 HARDWARE PRELIMINARIES

The choice of additive manufacturing technology constraints the subsequent numerical modeling. To keep the applicability of our developed system as wide as possible, we opted for a direct write

needle deposition system mounted on a 3-axis Cartesian robot (inset). The robot allows us to freely control the acceleration and position of the dispenser. The dispenser can process a wide range of viscous materials, and the deposition is very similar to fused deposition modeling. We further enhance the apparatus with two camera modules. The cameras lie on the opposite sides of the nozzle to allow our apparatus to perceive the location around the deposition. It is this locality of the in-situ view that we will leverage to formulate our numerical model.

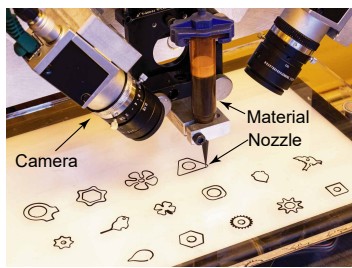

### 3.1 BASELINE CONTROLLER

To control the printing apparatus, we employ a baseline slicer. The input to the slicer is a three-dimensional object. The output is a series of locations the printing head visits to reproduce the model as closely as possible. To generate a single slice of the object, we start by intersecting the 3D model with a Z-axis aligned plane (please note that this does not affect the generalizability since the input can be arbitrarily rotated). The slice is represented by a polygon that marks the outline of the printout (Figure 1 gray). To generate the printing path, we assume a constant width of deposition (Figure 1 red) that acts as a convolution on the printing path. The printing path (Figure 1 blue) is created by offsetting the print boundary by half the width of the material using the Clipper algorithm (Johnson, 2015). The infill pattern is generated by tracing a zig-zag line through the area of the print (Figure 1 green).

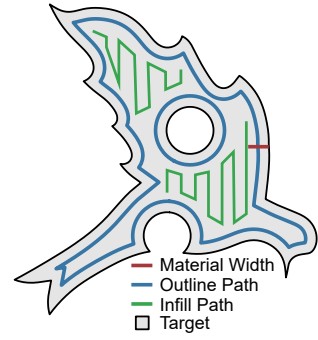

Figure 1: Baseline slicer.

## 4 REINFORCEMENT LEARNING FOR ADDITIVE MANUFACTURING

The baseline control strictly relies on a constant width of the material. To discover policies that can adapt to the in-situ observations, we formulate the search in a reinforcement learning framework. The control problem is described by a Markov decision process $(\mathcal{S}, \mathcal{A}, \mathcal{P}, \mathcal{R})$, where $\mathcal{S}$ is the observation space, $\mathcal{A}$ is a continuous action space, $\mathcal{P} = P(s'|s, a)$ is the transition function, and $\mathcal{R}(s, a) \to \mathbb{R}$ is the reward function.

To learn a control policy we take a model free approach by learning directly from printing. Unfortunately, learning on a physical device is challenging. The interaction between various process parameters can lead to deposition errors that require manual attention. As such discovering control policies directly on the hardware has too steep sample complexity to be practical. A potential solution is to learn the control behavior in simulation and transfer to the physical device. However, transfer from simulation to real world is a notoriously hard problem that hinges on applicability of the learned knowledge. In this work, we propose a framework for preparing numerical models for additive manufacturing that facilitate the sim-to-real transfer. Our model has three key components that facilitate the generalization of the learned control policies.

The first component is the design of the observation space. To facilitate the transfer of learning between simulation and a physical device, we rely on an abstraction of the observation space (Kaufmann et al., 2020). Rather than using the direct appearance feed from our camera module we process the signal into a heightmap. A heightmap is a 2D image where each pixel stores the height of the deposited material. For each height map location, the height is measured as a distance from the building plate to the deposited material. This allows our system to generalize to many different sensors such as cameras, depth sensors, or laser profilometers. However, unlike Kaufmann et al. (2020), we do not extract the feature vectors manually. Instead, similarly to OpenAI et al. (2018), we learn the features directly from the heightmap. In contrast to OpenAI et al. (2018), we do not randomize the observation domain. Additional randomization is not necessary in our case thanks to the controlled observation conditions of the physical apparatus.

A key insight of our approach is that the engineered observation space coupled with learned features can significantly help with policy learning. A careful design of the observation space can facilitate

the sim-to-real transfer, make the hardware design more flexible by enabling the use of a range of sensors that compute similar observations, and remove the need to hand-craft the features. It is therefore worth wile to invest in the design of observation spaces.

The second component of our system is the design of the action space. Instead of directly controlling the motors of the printer we rely on a high-level control scheme and tune coupled parameters such as velocity or offset from the printing path. This idea is similar in spirit to OpenAI et al. (2018). OpenAI et al. (2018) suggest not using direct sensory inputs from the mechanical hand as observations due to their noisiness and lack of generalization across environments. Instead, they use image data to track the robotic hand. Similarly, but instead in action space, we do not control the printer by directly inputting the typically noisy and hardware-specific voltages that actuate the motors of the apparatus. Instead, we control the printer by setting the desired velocity and offset and letting the apparatus match them to the best of its capabilities. This translation layer allows us to utilize the controller on a broader range of devices without per-device training.

This idea could also be generalized to other robotic tasks, for example, by applying a hierarchical divide and conquer approach to the action space. The control policies could output only high-level actions such as desired locations for robots actuators or deviations from a baseline behavior. Low-level controllers could then execute these higher-level actions. Such a control hierarchy can facilitate training by decoupling the higher-level goals from low-level inputs and transferring existing control policies to new devices through specialized low-level controllers.

The third and last component of our system is an approximative transition function. Rather than modelling the deposition process exactly we propose to approximate it qualitatively. A qualitative approximation allows us to design an efficient computational model. To facilitate the transfer of the simulated model to the physical device we reintroduce the device uncertainty in a data-driven fashion. This is similar to OpenAI et al. (2018), but instead of covering a large array of options, we specialize the randomization. Inspired by Chebotar et al. (2019), we designed a data-driven LPC filter that matches the statistical distribution of variations observed during a typical printing process. This noise enables our control policies to adapt to changing environments and, to some extent, to changes in material properties such as viscosity.

Our approximative transition function shows that it is not necessary to reproduce the physical world in simulation perfectly. A qualitative approximation is sufficient as long as we learn behavior patterns that translate to real-world experiences. This is an important observation for any task where we manipulate objects and elastic or frictional forces dominate the behavior. Relying on computationally more affordable simulations allows for applying existing learning algorithms to a broader range of problems where precise numerical modeling has prohibitive computational complexity. Moreover, by leveraging a numerical model it is possible to utilize privileged information that would be challenging if not impossible to collect in the real world. For full description of our methods please see Appendix A.

## 5 RESULTS

In this section, we provide results obtained in both virtual and physical environments. We first show that an adaptive policy can outperform baseline approaches in environments with constant deposition. Next, we showcase the in-process monitoring and the ability of our policy to adapt to dynamic environments. Finally, we demonstrate our learned controllers transferring to the physical world with a minimal sim-to-real gap.

### 5.1 COMPARISON WITH BASELINE CONTROLLER

We evaluate the optimized control scheme on a selection of freeform and CAD models sampled from Thingy10k (Zhou & Jacobson, 2016) and ABC (Koch et al., 2019) datasets (Appendix A.6). In total, we have 113 unseen slices corresponding to 96 unseen geometries. We report our findings in Figure 2. For each input slice, we report improvement on the printed boundary as the average offset. The average offset is defined as a sum of areas of under and over deposited material normalized by the outline length. More specifically, given an image of the target slice $T$, printed canvas $C$, a weight

mask $W$, and the length of the outline $l$, the average offset $\mathcal{O}$ is computed as:

$$\mathcal{O} = \frac{(1 - C)TW}{l} + \frac{C(1 - T)}{l}. \tag{1}$$

The improvement is calculated as a difference between the baseline and our policy. Therefore, a value higher than zero indicates that our control policy outperformed the baseline. As we can see, our policy achieved better performance in all considered models.

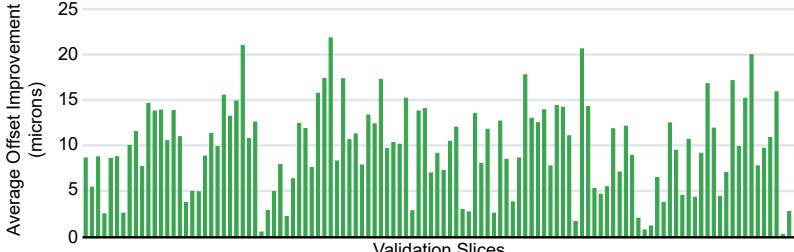

Figure 2: The relative improvement of our policy over baseline.

Next, we investigate the shapes where our control policy achieves the highest and the lowest gain, respectively (Figure 3). Best performance is achieved in smooth regions. The reason is that our policy is capable of adjusting the printing parameters based on the curvature while the baseline's constant speed is more suitable for a limited range of curvatures. Conversely, our policy achieves the weakest performance on objects with sharp features. This is natural as the width of the deposited material in sharp regions is too large for the desired feature scale, leading to over-deposition. If such thin features are desired to print regularly, a thinner material nozzle can alleviate this issue.

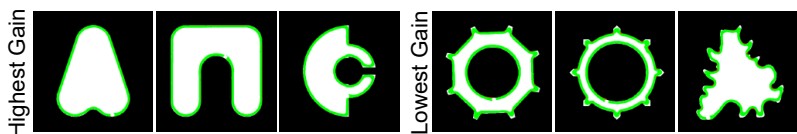

Figure 3: Representative deposited patterns from the evaluation dataset.

Finally, we compare our control policy with fine-tuned baseline. The baseline controller uses the same parameters for each slice. Different process parameters may be optimal for different slices. To this end, we choose two slices, a freeform slice of a *bird* and a CAD slice of a *bolt* and optimize their process parameters using Bayesian optimization, Figure 26 (for numerical details see Appendix B). We can observe that the two control schemes require drastically different velocities (1.46 SU/s vs. 0.89 SU/s) to maximize performance. Moreover, we can see that the control parameters are not interchangeable, Appendix B. When switching the control parameters, we can observe a loss in performance. This loss is caused by each set of control parameters exploiting the local properties of the slice used for training. Lastly, we compare the individually optimized control parameters with our policy. Our policy improves upon both baseline solutions while maintaining generalizability. This is possible because our control policy relies on live feedback to adjust the printing parameters on-the-fly.

### 5.1.1 ABLATION STUDY ON OBSERVATION SPACE

Our control policy relies on a live view of the deposition system to select the control parameters. However, the in-situ view is a technologically challenging addition to the printer hardware that requires a carefully calibrated imagining setup. With this ablation study, we verify how important the individual observations are to the final print quality. We consider three cases: (1) no printing bed view, (2) no target view, and (3) no future path view. We analyzed the results from the pre-test (full observation space $\mu = 9.7$, $\sigma = 4.9$) and the post-tests (no canvas $\mu = 8.8$, $\sigma = 5.7$, no target $\mu = 7.2$, $\sigma = 5.5$, no path $\mu = 8.4$, $\sigma = 4.8$) printing task using paired t-tests with Holm-Bonferroni correction. The analysis indicates that the availability of all three inputs: the printing bed, the target, and the path improved final printouts ($P < 0.01$ for all three cases).

## 5.2 PERFORMANCE IN DYNAMIC ENVIRONMENTS

We use an identical random pressure variation profile to perform a quantitative evaluation in environments with varying pressure. We use the same evaluation dataset as for constant-pressure policies and report the overall improvement over the baseline controller, (Figure 4). We can observe that in each of the considered slices, our closed-loop controller outperformed the baseline.

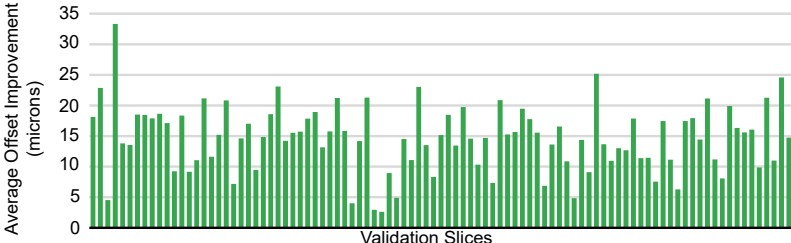

Figure 4: The relative improvement of our policy over baseline.

We have also evaluated the infill policy in a noisy environment, (Figure 5). We can observe that the deposition noise leads to an accumulation of material. The accumulation eventually results in a bulge of material in the center of the print, complicating the deposition of subsequent layers as the material would tend to slide off. In contrast, our policy dynamically adjusts the printing path to generate a print with significantly better height uniformity. As we can observe, the surface generated by our policy is almost flat and ready for deposition of potentially more layers.

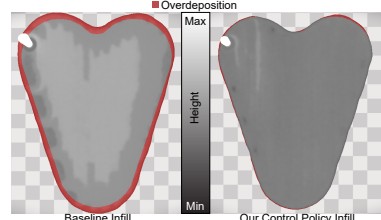

Figure 5: Infill comparison.

## 5.3 ABLATION STUDY ON VISCOSITY

To verify that our policy can adapt to various materials, we trained three models of varying viscosity, (Figure 6). We can observe that, without an adaptive control scheme, the pressure changes are sufficiently strong to cause local over- or under-deposition. Our trained policy dynamically adjusts the offset and velocity to counterbalance the changes in the deposition. We can see that our policy is particularly good at handling smooth width changes and quickly recovers from a spike in printing width.

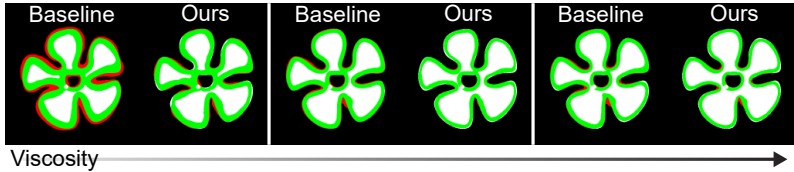

Figure 6: Performance of our policy and baseline with varying viscosity.

### 5.3.1 ABLATION STUDY ON ACTION SPACE

To evaluate the need to tweak both the printing velocity and the printing path, we trained two control policies with a limited action set to either alter the velocity or path offset. We analyzed the results from the pre-test (full action space $\mu = 12.7$, $\sigma = 5.7$) and the post-tests (velocity $\mu = 7.5$, $\sigma = 2.5$, displacement $\mu = 5.6$, $\sigma = 8.3$) printing task using paired t-

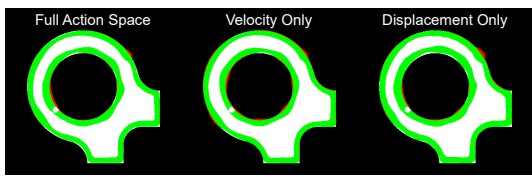

Figure 7: Action space ablation study.

tests with Holm-Bonferroni correction. The analysis indicates that the availability of the full action space resulted in an improvement in final printouts ($P < 0.001$ for both cases). The difference in performance depends on the inherent limitations of the individual actions. On the one hand, adjusting velocity is fast (under 6.6 milliseconds) but can cope only with moderate changes in material

width. This can be observed as the larger bulges of over-deposited material in Figure 7 middle. On the other hand, while offset can cope with larger material differences but it needs between 0.13 and 1.3 seconds to adjust. As a result, offset adjustment cannot cope with sudden material changes, (Figure 7 right). In contrast, by utilizing the full action space our policy can combine the advantages of the individual actions and minimize over-deposition, (Figure 7 left).

### 5.3.2 ABLATION STUDY ON REWARD FUNCTION

Our reward function uses privileged information from the numerical simulation to evaluate how material settles over time. However, such information is not readily available on physical hardware. One either evaluates the reward once at the end of each episode to include material flow or at each timestep by disregarding long-term material motion. We evaluated how such changes to the reward function would affect our control policies. We analyzed the results from the pre-test (privileged reward $\mu = 12.7$, $\sigma = 5.7$) and the post-tests (delayed reward $\mu = -22.3$, $\sigma = 8.6$, immediate reward $\mu = 9.2$, $\sigma = 8.0$) printing task using paired t-tests with Holm-Bonferroni correction. The analysis indicates that the availability of the privileged information resulted in an improvement in final patterns ($P < 0.001$ for both cases). The learning process for a delayed reward is significantly slower, and it is unclear if performance similar to our policy can be achieved, Appendix A.6. On the other hand, the immediate reward policy learns faster but cannot handle material changes over longer time horizons, (Figure 8).

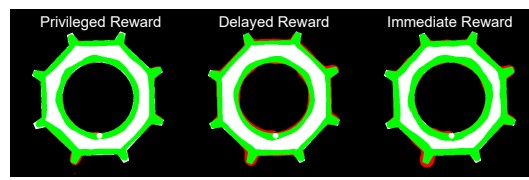

Figure 8: Reward function ablation study.

### 5.4 PERFORMANCE ON PHYSICAL HARDWARE

Finally, we evaluate our control policies on physical hardware. The policies were trained entirely in simulation without any additional fine-tuning on the printing device. To conduct the evaluation, we equipped our printer with a pressure controller. The pressure control was set to a sinusoidal oscillatory signal to provide a controllable dynamic change in material properties. We used two materials, with high and low viscosity, and used two separate policies pretrained in simulation using those materials. We printed 22 slices, of which 11 corresponded to the simulation training set and 11 to the evaluation set. We monitor the printing process and use the captured images to run our evaluation function to capture quantitative results. We observe that our controllers improve the average offset over the baseline print in every scenario, (Figure 9).

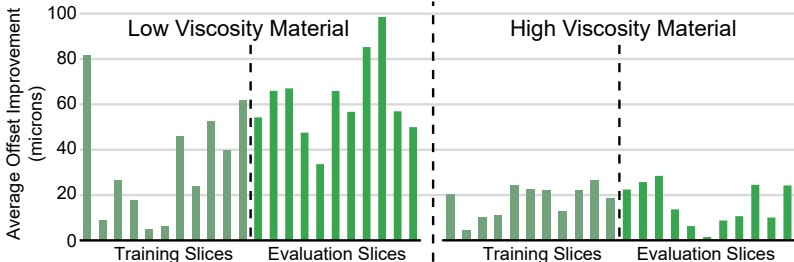

Figure 9: The relative improvement of our policy over baseline.

A sample of the fabricated slices can be seen in Figure 10. The print target (white) is overlaid with a map of underdeposited (blue) and overdeposited (red) material. We further plot a histogram of under and over deposition.

We can see that our control policy transferred excellently to the physical hardware without any additional training. Our policy consistently achieves smaller over-deposition while not suffering from significant under-deposition. Moreover, in many cases our policy achieves histograms with smaller width suggesting we achieved a tighter control over the material deposition than the baseline. This demonstrates that our numerical model enables learning control policies for additive manufacturing in simulation.

# 6 CONCLUSION

We present the first closed-loop control policy for additive manufacturing recovered via reinforcement learning. To learn an effective control policy, we propose a custom numerical model of the deposition process. During the design of our model, we tackle several challenges. To obtain an efficient approximation of the deposition process, we leverage the limited perception of a printing apparatus and model the deposition only qualitatively. To include non-linear coupling between process parameters and printed materials, we utilize a data-driven predictive model for the deposition imperfections. Finally, to enable long horizon learning with viscous materials, we use the privileged information generated by our numerical model for reward computation. In several ablation studies, we show that these components are required to achieve high-quality printing, effectively react to instantaneous and long horizon material changes, handle materials with varying viscosity, and adapt the deposition parameters to achieve printouts with minimal over-deposition and smooth top layers.

We demonstrate that our model can be used to train control policies that outperform baseline controllers, and transfer to physical apparatus with a minimal sim-to-real gap. We showcase this by applying control policies trained exclusively in simulation on a physical printing apparatus. We use our policies to fabricate several prototypes using low and high viscosity materials. The quantitative and qualitative analysis clearly shows the improvement of our controllers over baseline printing. This indicates that our numerical model can guide the future development of closed-loop policies for additive manufacturing. Thanks to its minimal sim-to-real gap, the model democratizes research on learning for additive manufacturing by limiting the need to invest in specialized hardware. Furthermore, by expanding the simulator with other physical phenomena, e.g., abrasion, melting, or heat transfer, our numerical model can serve as a blueprint for learning closed-loop control policies of other manufacturing methods such as milling, direct energy deposition, or selective laser sintering.

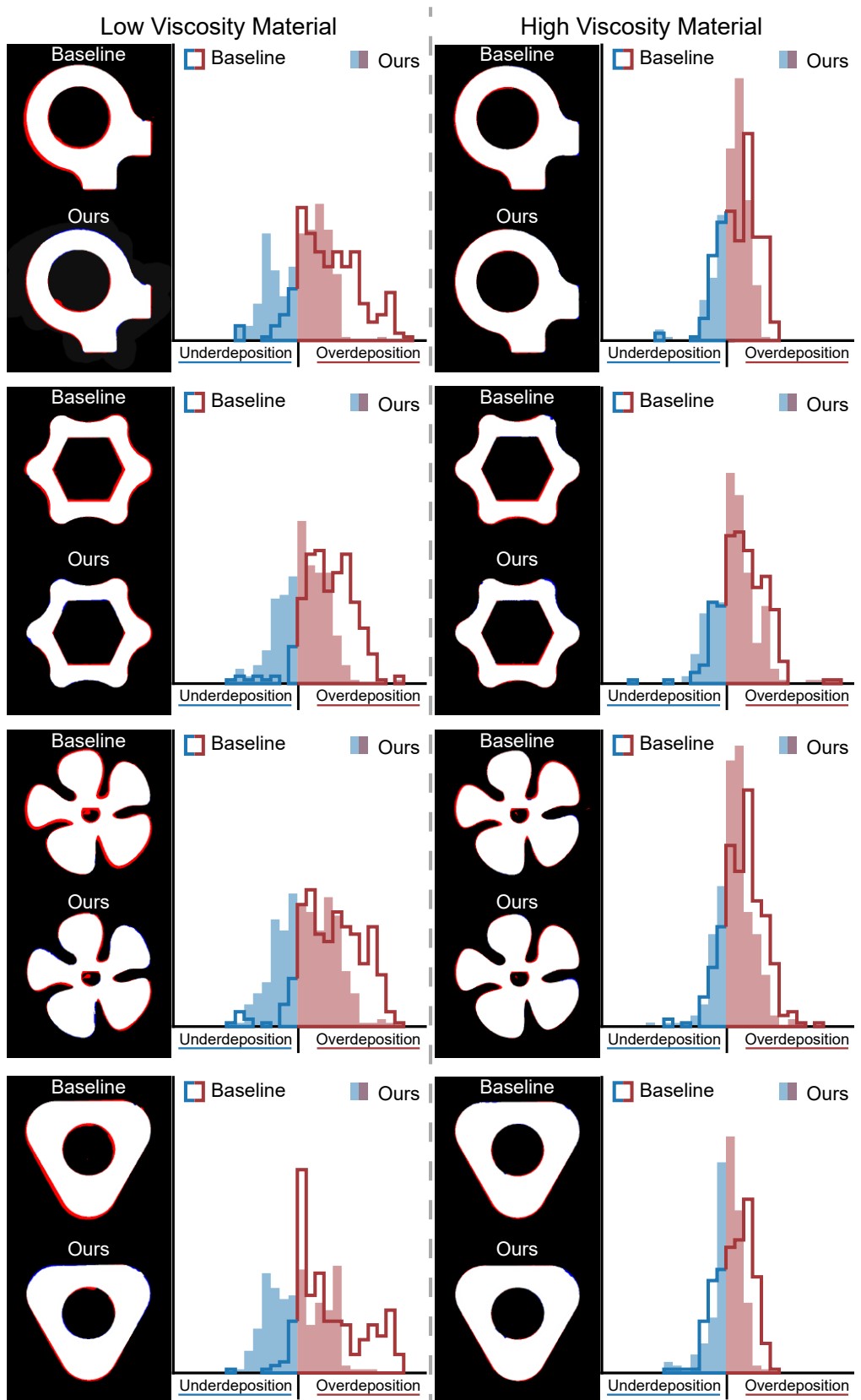

Figure 10: Deposition quality estimation of physical result manufactured with baseline and our learned policy.

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

## A  Methods

### A.1  Hardware Setup

In this work, we developed a direct write 3D printing platform with an optical feedback system that can measure the dispensed material real-time, in-situ. The 3D printer is comprised of a pressure driven syringe pump and pressure controller, a 3-axis Cartesian robot, optical imaging system, back-lit build platform, 3D-printer controller and CPU Figure 11. The 3-axis Cartesian robot is used to locate the build platform in x and y-direction and the print carriage in the z-direction. The pressure driven syringe pump and pressure controller are used to dispense and optically opaque material onto the back-lit build platform. The back-lit platform is used to illuminate the dispensed material. The movement of the robot, actuation of the syringe pump and timing of the cameras are controlled via the controller. The CPU is used to process the images after they are acquired and compute updated commands to send to the controller.

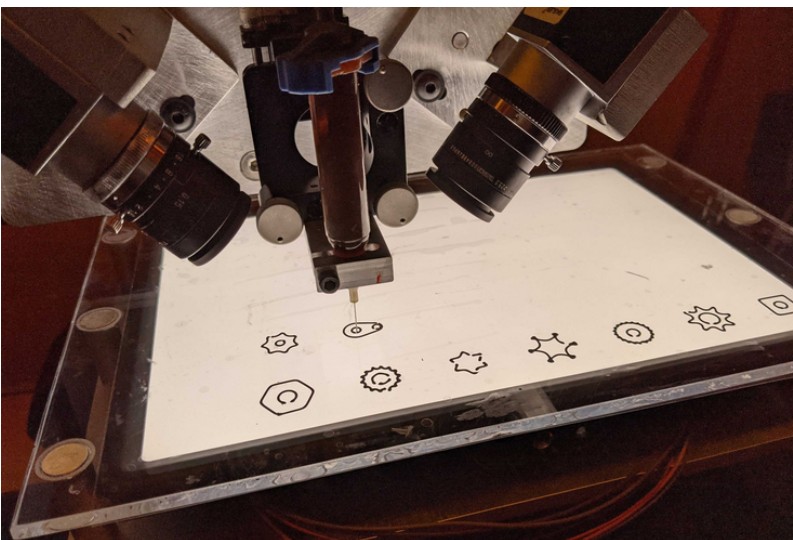

Figure 11: The printing apparatus consisting of a 3-axis Cartesian robot, a direct write printing head, and a camera setup.

### A.1.1  Calibration

To enable realtime control of the printing process we implemented an in-situ view of the material deposition. Ideally we would capture a top-down view of the deposited material. Unfortunately, this is not possible since the material is obstructed by the dispensing nozzle. As a result the camera has to observe the printing bed from an angle. Since the nozzle would obstruct the view of any single camera we opted to use two cameras. More specifically, we place two CMOS cameras (Basler AG, Ahrensburg, Germany) at 45 degrees on each side of the dispensing nozzle, Figure 11. We calibrate the camera by collecting a set of images and estimating its intrinsic parameters, Figure 12 calibration. To obtain a single top-down view we capture a calibration target aligned with the image frames of both cameras, Figure 12 homography. By calculating the homography between the captured targets and an ideal top-down view we can stitch the images into a single view from a virtual over-the-top camera. Finally, we mask the location of each nozzle in the image (Figure 12 nozzle masks) and obtain the final in-situ view, Figure 12 stitched image.

The recovered in-situ view is scaled to attain the same universal scene unit size as our control policies are trained in. Since we seek to model the deposition only qualitatively it is sufficient to rescale the in-situ view to match the scale of the virtual environments. We identify this scaling factor separately for each material. To calibrate a single material we start by depositing a straight line at maximum velocity. The scaling factor is then the ratio required to match the observed thickness of the line with simulation. To extract the thickness of the deposited material we rely on its translucency properties. More precisely, we correlate material thickness with optical intensity. We do this be depositing the

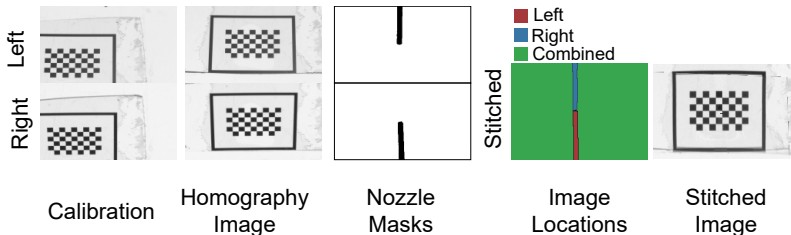

Figure 12: The calibration of the imaging setup. First intrinsic parameters are estimated from calibration patterns. Next we compute the extrinsic calibration by calculating homographies between the cameras and an overhead view. We extract the masks by thresholding a photo of the nozzle. The final stitched image consists of 4 regions: (1) view only in left camera, (2) view only in right camera, (3) view in both cameras, (4) view in no camera. The final stitched image is show on the right.

material at various thickness and taking a picture with our camera setup. The optical intensity then decays exponentially with increased thickness which is captured by a power law fit.

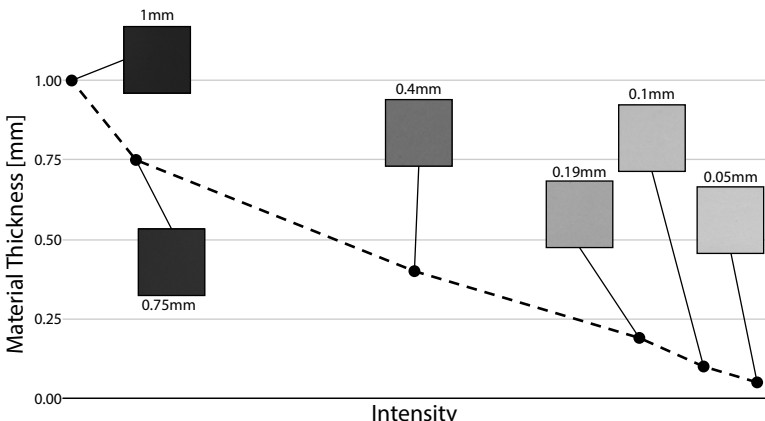

Figure 13: Calibration images for correlating deposited material thickness with optical intensity and the corresponding fit.

The last assumption of our control policy is that the deposition needle is centered with respect to the in-situ view. To ensure that this assumption holds with the physical hardware we calibrate the location of the dispensing needle within the field of view of each camera and with respect to the build platform. First, a dial indicator is used to measure the height of the nozzle in z and the fine adjustment stage is adjusted until the nozzle is 254 microns above the print platform. Next, using a calibration target located on the build platform and the fine adjustment stage, the nozzle is centered in the field of view of each camera. This calibration procedure is done each time the nozzle is replaced during the start of each printing session.

### A.1.2 BASELINE CONTROLLER

To calibrate the baseline control we follow the same procedure in simulation and physical hardware. We start be depositing a straight line at a constant velocity, Figure 14. Next, we measure the width of the deposited line at various locations to estimate the mean width. We use the width to generate the offset for outline printing and spacing of the infill pattern.

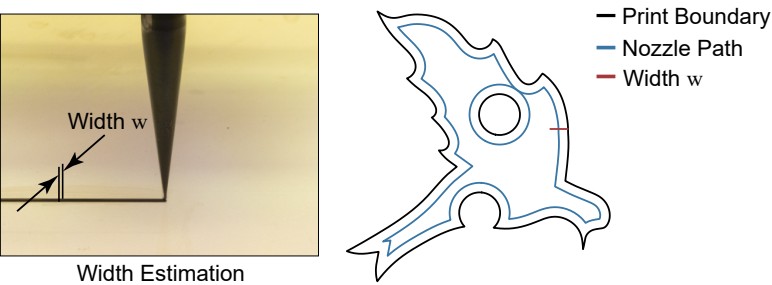

Figure 14: Baseline controller starts by estimating the width $w$ of the deposited material. A control sequence for the nozzle is estimated by offsetting the desired shape by half the size of material width.

## A.2 CONTROL POLICY INPUT STATES

To define the input states, we closely follow the constraints of the physical hardware. We model our observation space as a small in-situ view centered at the printing nozzle location. The view has a size of $84 \times 84$ pixels which translate to roughly $2.95 \times 2.95$ scene units (SU). The view contains either a heightmap (for infill printing) or material segmentation (for outline printing). Since the location directly under the nozzle is obscured for the physical hardware,

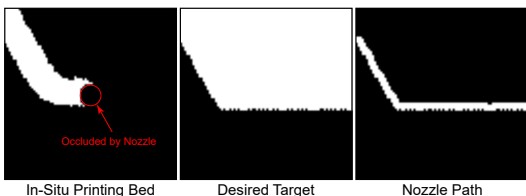

Figure 15: Control Policy Input.

we mask a small central position in the view equivalent to $0.42$ SU or $\frac{1}{7}$th of the in-situ view. Together with the local view, we also provide the printer with a local image of the desired printing target and the path the control policy will take in the environment. To further minimize data redundancy, we rotate the in-situ view such that the printer moves along the positive X-axis in the image. These three inputs are stacked together into a 3-channel image, (Figure 15).

## A.3 ACTION SPACE

The selection of action space plays a critical role in adapting a controller to the actual hardware. One possibility is to control and directly modify the voltage input of individual motors. However, such an approach is not readily transferable between printing devices. The controls are tied too tightly to the hardware selection and would exaggerate the sim-to-real gap. Moreover, directly affecting the motor voltage would mean that the control policy must learn how to trace print inputs. Instead, we propose a strategy that leverages the body of work on designing baseline controllers. Similar to baseline, our control policy follows a path generated by a slicer. However, we enable dynamic modification of the path. At each state, the printer can modify two actions: (1) the velocity at which the printing head is moving and (2) displacement of the printing head in a direction perpendicular to the motion, Figure 16. Such a formulation allows us to decouple the hardware parameters from the control scheme and apply the same policy in both simulation and physical hardware by

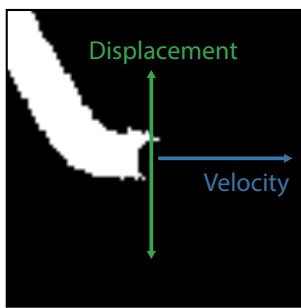

Figure 16: The action space.

scaling the input units appropriately. In our simulation, we limit our velocity to the range of $[0.2, 2]$ SU/s and the displacement to $0.2666$ SU.

## A.4 TRANSITION FUNCTION

The transition function takes a state-action pair and outputs a new state of the environment. In our setting, this means we need to numerically model the fabrication process, which is a notoriously

difficult problem. Here we leverage our assumption that the observation space is so localized that it can identify the deposited materials only qualitatively. Therefore, we can trade physical realism for visual fidelity and efficiency. This description fits the Position-Based-Dynamics (PBD) (Macklin & Müller, 2013) framework, which is a geometrical approximation to the equations of motion.

To model the interaction of the deposited material with the printing apparatus we rely on Position-Based Dynamics (PBD). PBD approximates rigid, viscous, and fluid objects as collections of particles. To represent the fluid we assume a set of $N$ particles where each particle is defined by its position $\mathbf{p}$, velocity $\mathbf{v}$, mass $m$, and a set of constraints $C$. In our setting we consider two constraints: (1) collision with the nozzle and (2) incompressibility of the fluid material. We model the collision with the nozzle as a hard inequality constraint:

$$C_i(\mathbf{p_i}) = (\mathbf{p_i} - \mathbf{q}_c) \cdot \mathbf{n}_c, \tag{2}$$

where $\mathbf{q}_c$ is the contact point of a particle with the nozzle geometry along the direction of particles motion $\mathbf{v}$ and $\mathbf{n}_c$ is the normal at the contact location. To ensure that our fluids remain incompressilbe we follow (Macklin & Müller, 2013) and formulate a density constraint for each particle:

$$C_i(\mathbf{p}_1, ..., \mathbf{p}_n) = \frac{\rho_i}{\rho_0} - 1, \tag{3}$$

$$\rho_i = \sum_j m_j W(\mathbf{p}_i - \mathbf{p}_j, h), \tag{4}$$

where $\rho_0$ is the rest density and $\rho_i$ is given by a Smoothed Particle Hydrodynamics estimator (Müller et al., 2003) in which $W$ is the smoothing kernel defined by the smoothing scale $h$.

We further tune the simulation parameters to achieve a wide range of viscosity properties. More specifically, we couple the effects of viscosity, adhesion, and energy dissipation into a single setting. By coupling these parameters we obtain materials with optically different viscosity properties. Moreover, we noticed that the number of solving substeps has a significant effect on viscosity and surface tension of the simulated fluids. Therefore, we also tweak the number of substeps from 2 for liquid-like materials to 5 for highly-viscous materials.

We replicate our printing apparatus in the simulation, inset. We model the nozzle as a collision object with a hard contact constraint on the fluid particles. Since modeling a pressurized reservoir is computationally costly as it requires us to have many particles in constant contact, we chose to approximate the deposition process at the peak of the nozzle. More specifically, we model the deposition as a particle emitter. To set the volume and velocity of the particles, we use a flow setting. The higher the flow, the more particles with higher initial velocities are generated. This qualitatively approximates the deposition process with a pressurized reservoir.

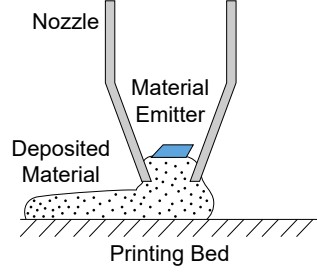

The particle emitter is placed slightly inside the nozzle to allow for realistic material buildup and a delayed stop, similar to extrusion processes. Finally, we consider the printer to have only a finite acceleration per timestep. To accelerate to target velocity, we employ a linear acceleration scheme.

Another important choice for the numerical model is the used discretization. We have two options: (1) time-based and (2) distance-based. We originally experimented with time-based discretization. However, we found out that time discretization is not suitable for printer modeling. As the velocity in simulation approaches zero, the difference in deposited material becomes progressively smaller until the gradient information completely vanishes, Figure 17 left. Moreover, a time-based discretization allows the policy to affect the number of evaluations of the environment directly. As a result, it can avoid being punished for bad material deposition by quickly rushing the environment to finish. Considering these factors we opted for distance-based discretization, Figure 17 right. The policy specifies the desired velocity at each interaction point, and the environment travels a predefined distance (0.2666 SU) at the desired speed. This helps to regularize the reward function and enable learning of varying control policies.

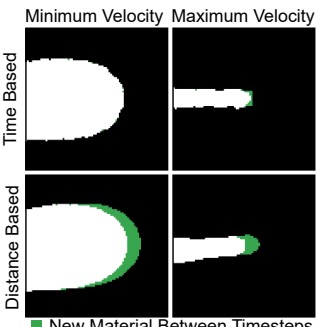

Figure 17: Discretization.

An interesting design element is the orientation of the control polygons created by the slicer. When the outline is defined as points given counter-clockwise, then due to the applied rotation, each view is split roughly into two half-spaces, (Figure 18). The bottom one corresponds to outside i.e., generally black, and the upper one corresponds to inside i.e., generally white. However, the situation changes when outlining a hole. When printing a hole the two half-spaces swap location. We can remove this disambiguity by changing the orientation of the polylines defining holes in the model. By orienting them clockwise, we will effectively swap the two half-spaces to the same orientation as when printing the outer part. As a result, we achieve a better usage of trajectories and a more robust control scheme that does not need to be separately trained for each print's outer and inner parts.

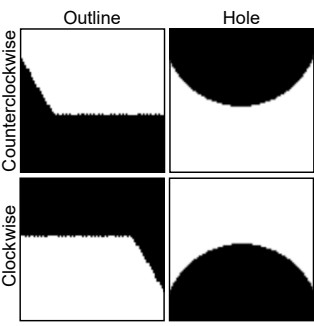

Figure 18: Orientation.

To design a realistic virtual printing environment, the model needs to capture the deposition imperfections. The source of these imperfections is the complex non-linear coupling between the dynamic material properties and the deposition parameters. Analytical modeling of this coupling is challenging as it requires a deep understanding of these interactions. Instead, we adopted a data-driven model. We observe that the final effect of the deposition error is a varying width of the deposited material. To recover such a model for our apparatus, we start by printing a reference slice over multiple iterations, (Figure 19 left). At each iteration, we measure the width of the deposited material at specified cross-sections, (Figure 19 middle). This yields us observations of how the material width evolves in time, (Figure 19 left). To formulate a predictive generative model, we employ a tool from speech processing called Linear Predictive Coding (LPC) (Marple, 1980). The model assumes that a signal is generated by a buzz filtered by an auto-correlation filter. We use this assumption to recover filter coefficients that transform white Gaussian noise into realistic pressure samples, (Figure 19 left).

To formulate a predictive generative model we employ a tool from speech processing called Linear Predictive Coding (LPC) (Marple, 1980). We can predict the next sample of a signal as a weighted sum of $M$ past output samples and a noise term:

$$x_n = -\sum_{m=1}^{M} a_{M,m} x_{n-m} + \epsilon_n, \tag{5}$$

where $x$ are the signal samples, $\epsilon$ is the noise term, and $a_{M,m}$ are the parameters of $M$-th order auto-correlation filter. To find these coefficients Burg (1975) propose to minimize the following energies:

$$e_M = \sum_{k=1}^{N-m} |f_{M,k}|^2 + \sum_{k=1}^{N-m} |b_{M,k}|^2, \tag{6}$$

$$f_{M,k} = \sum_{i=0}^{M} a_{M,i} x_{k+M-i}, \tag{7}$$

$$b_{M,k} = \sum_{i=0}^{M} a_{M,i}^* x_{k+i}, \tag{8}$$

where $*$ denotes the complex conjugate. After finding the filter coefficients with Equation 6 we can synthesize new width variations with similar frequency composition to the physical hardware by filtering a buzz modeled as a white Gaussian noise. Since we sampled the width variation at discrete intervals we further find a smooth interpolating curve that corresponds model the observed pressure variation. We use the proposed model to drive the flow setting of our simulator. This directly influences the width of the deposited material similarly to the imperfections in the deposition.

### A.5 REWARD FUNCTION

Viscous materials take significant time to settle after deposition. Therefore, to assess deposition errors, it is needed to observe the deposition over long horizons. However, the localized nature

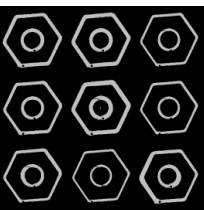 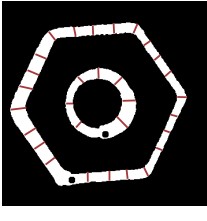 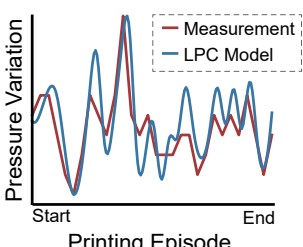

Calibration Printouts     Sample Locations     Printing Episode

Figure 19: We performed nine printouts and measured the width variation at specified locations. We fit the measured data with an LPC model. Please note that since our model is generative, we do not exactly match the data or any observed resemblance is a testament to the quality of our predictor.

of the in-situ view makes such observations impossible on the physical hardware. As a result, learning long-horizon planning has infeasible sample complexity. To tackle this issue, we leverage the fact that we utilize a numerical approximation of the deposition process with access to privileged information. At each simulation step, we model the entire printing bed. This allows us to formulate the reward function as a global print quality metric. More specifically, our metric is composed of two terms: (1) a reward term for depositing material inside the desired slice and (2) a punishment term for depositing material outside of the slice. To keep the values consistent across slices of varying size, we normalize them by the length of the outline or the infill area, respectively. We provide dense rewards as the difference between the metrics evaluated at two subsequent timesteps to accelerate the training further.

We consider two reward function in our setting one for outline printing and one for infill printing. Each reward function evaluates the print quality as a whole. To accelerate the learning we provide the algorithm with dense rewards as a delta between the reward in-between steps $R = R^{n+1} - R^n$.

To print the outline we want to follow the boundary as closely as possible without overfilling. To this end we compose our reward function of two terms. Given an image of the current printing bed $C$ and the desired target $T$ we define the reward as $\sum CT$. While such a formulation rewards the control policy for depositing material inside the printing volume it does not encourage a tight outline fill. Indeed a potential strategy with such a reward would be to offset the printing nozzle as much inside as possible and then move safely within the object bounds. To address this issue we propose to include a weight map $W$ that is computed as a thresholded distance transform of the target $T$. The final reward function is then: $R = \sum CTW$. Using such a formulation we put the highest weight on depositing directly on the outline boundary. The threshold cutoff then helps preventing a strategy of filling up the shape interior. To ensure that the printer deposits material inside the desired locations we include an additional punishment term $P = \sum C(1 - T)$. Finally, both reward and punishment is normalized by the length of the outline of our target.

For infill printing we compute the reward from the heightfield of the deposited material. We start by estimating how much of the slice was covered. To this end, we use a thresholded version of the canvas and compute the coverage as $R = \sum CT$. Similarly, we estimate the amount of over-deposited material as $P = \sum C(1 - T)$. To keep these values consistent across different slices we normalize them by the total area of the print. Finally, to motivate deposition of flat surfaces suitable for 3D printing we add another penalty term as the standard deviation of the canvas heightfield.

A.6    TRAINING PROCEDURE

To train our control policy we start with g-code generated by a slicer. As inputs to the slicer we consider a set of 3D models collected from the Thingy10k dataset. To train a controller the input models need to be carefully selected. On the one hand, if we pick an object with too low frequency features with respect to the printing nozzle size then any printing errors due to control policy will have negligible influence on the final result. On the other hand, if we pick a model with too high frequency features with respect to the printing nozzle then the nozzle will be physically unable to reproduce these features. As a result we opted for a manual selection of 18 models that span a wide variety of features, Figure 21. Each model is scaled to fit into a printing volume of $18 \times 18$ SU and sliced at random locations.

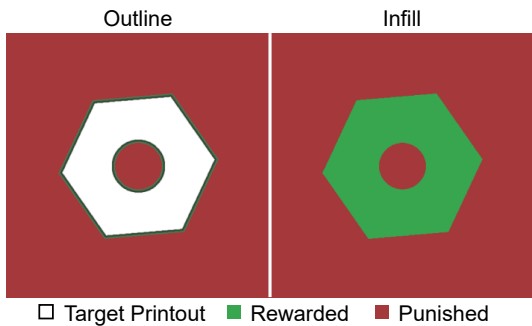

Figure 20: The reward function.

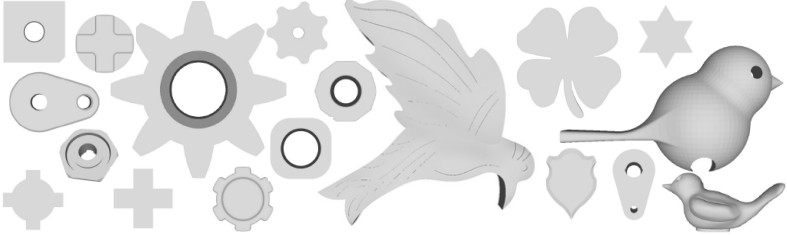

Figure 21: Models in our curriculum. For a full view of exemplar slices please see the supplementary material.

Our policy is represented as a CNN modeled after Mnih et al. (2015). The network imput is a $84 \times 84 \times 3$ image. The image is passed through three hidden layers. The convolution layers have the respective parameters: (32 filters, filter size 8, stride 4), (64 filters, filter size 4, stride 2), and (64 filters, filter size 3, stride 1). The final convolved image is linearized and passed through a fully-connected layer with 512 neurons that is connected to the output action. Each hidden layer uses the nonlinear rectifier activation. We formulate our objective function as:

$$\arg \max_{\theta} \mathbb{E}_t^{\mathcal{C}} \left[ \frac{\pi_{\theta_t}(a_t|s_t)}{\pi_{\theta_{t-1}}(a_t|s_t)} \hat{A}_t \right], \tag{9}$$

where $t$ is a timestep in the optimization, $\theta$ are the hyperparameters of a neural network encoding our policy $\pi$ that generates an action $a_t$ based on a set of observations $s_t$, $\hat{A}_t$ is the estimator of the advantage function and the expectation $\mathbb{E}_t^{\mathcal{C}}$ is an average of a finite batch of samples generated by printing sliced models from our curriculum $\mathcal{C}$. To maximize Equation 9 we use PPO algorithm (Schulman et al., 2017). Each trajectory consists of a randomly selected mesh slice that is fully printed out before proceeding to the next one. One epoch terminates when we collect 10000 observations. We run the algorithm for a total of 4 million observations but convergence was achieved well before that, Figure 22. For the training parameters we set the entropy coefficient to 0.01 and anneal it towards 0. Similarly we anneal the learning rate from 3e-4 towards zero. Lastly, we picked a discount factor of 0.99 which corresponds to one action having a half time of 70 steps. This is equivalent to roughly 18.6 SU of distance traveled. In our training set this corresponds to 29-80 percent of the total episode length.

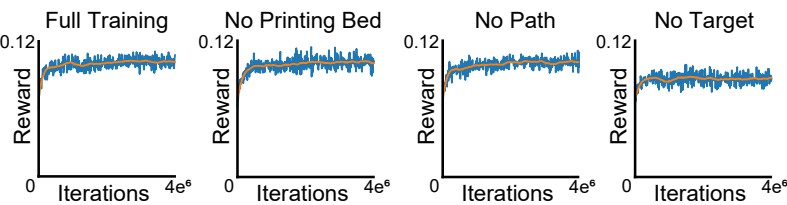

Figure 22: Training curves for controllers with constant material flow.

We also experimented with training controllers for materials with varying viscosity, Figure 23. In general we have observed that the change in viscosity did not significantly affect the learning convergence. However, we have observed a drop in performance when training control policies for deposition of liquid materials. The liquid material requires longer time horizons to stabilize and has a wider deposition area making precise tracing of fine features challenging.

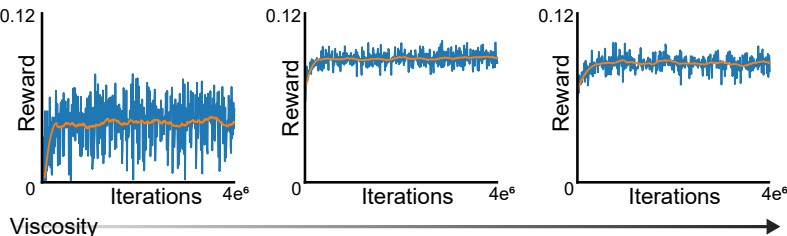

Figure 23: Training curves for controllers with increasing viscosity in an environment with noisy flow.

Lastly, we conducted ablation studies on action space and reward function in the environment with noisy deposition, Figure 24. We can see that employing the delayed reward had a negative effect on convergence and it is unclear if a policy of sufficient quality would be achieved.

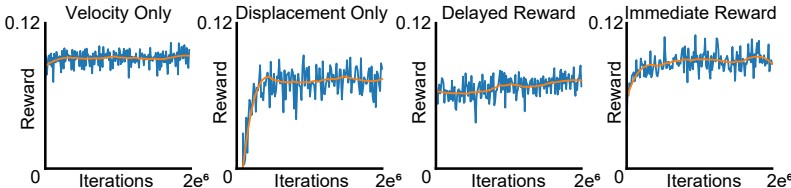

Figure 24: Training curves for controllers with variable material flow.

For evaluation we constructed a separate dataset consisting of freeform and CAD geometries that were not present in the training, Figure 25.

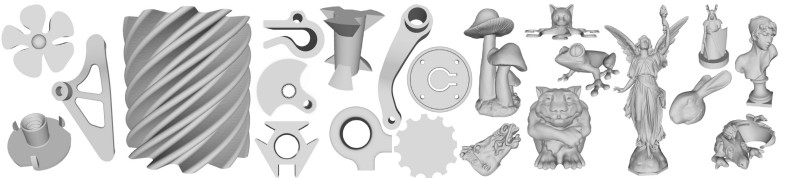

Figure 25: Exemplar models from the evaluation dataset.

## B  BAYESIAN OPTIMIZATION FOR BASELINE CONTROL

While the baseline controller closely follows the printed boundaries is possible that there is a more suitable policy to maximize our objective function. To verify this we use the environment described in Section 4 to search for a velocity and offset that maximizes the reward function. More specifically we optimize a simplified objective of Equation 9 limited to a single shape:

$$\arg\max_{v,d} \mathbb{E}\left[\pi_{v,d}(a_t|s_t)\right],\tag{10}$$

where $v$ and $d$ are the optimized velocity and displacement of the printing policy $\pi_{v,d}$, and $\mathbb{E}$ reduces to the expected cumulative reward of executing our proposed environment with a single slice. Maximizing Equation 10 even for a single shape is a challenging task due to the high cost associated with evaluating the objective function. Because of this we rely on Bayesian optimization to maximize the objective. We warm-start the optimization with 20 samples acquired through Latin sampling of

our 2-dimensional action space. We run the optimization until convergence that we define as not improving upon the found maxima for over 300 iterations. We can see the optimized controllers for a free-form *bird* model and a CAD model of a *bolt* compared to our optimized policy in Figure 26.

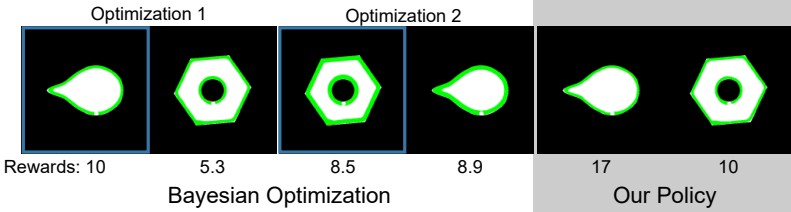

Figure 26: Printouts realized using control policies recovered with Bayesian optimization (left and middle, blue square marks the optimized slice) compared to our trained policy (right).

## C  ADAPTATION TO VARYING VISCOSITY

We evaluate how our learned controllers adapt to varying viscosity, (Figure 27). We can observe that our policy learned on low-viscosity materials consistently under-deposits when used to print at higher viscosities. Conversely, our control policy learned on high-viscosity material over-deposits when applied to materials with lower viscosities. From this observation we conclude that our policy learns the spread of the material post-deposition and uses this information to guide the deposition. Therefore, small viscosity variations are not likely to pose significat challenge for our learned policies. However, if the learned material behavior is significantly violated the in-situ observation space limits the ability of our policy to adapt to a before unseen material.

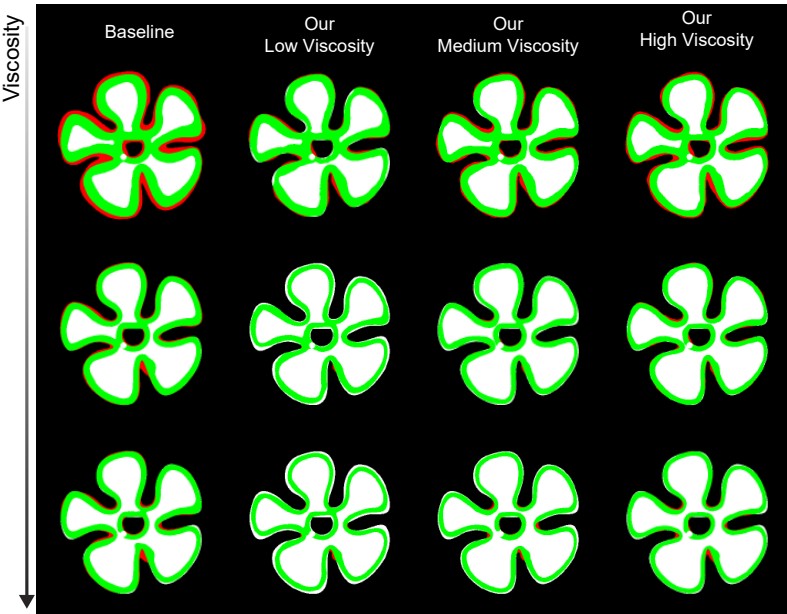

Figure 27: We compare the baseline policy and our three learned policies on materials with varying viscosity.

# D DETAILED PHYSICAL RESULTS

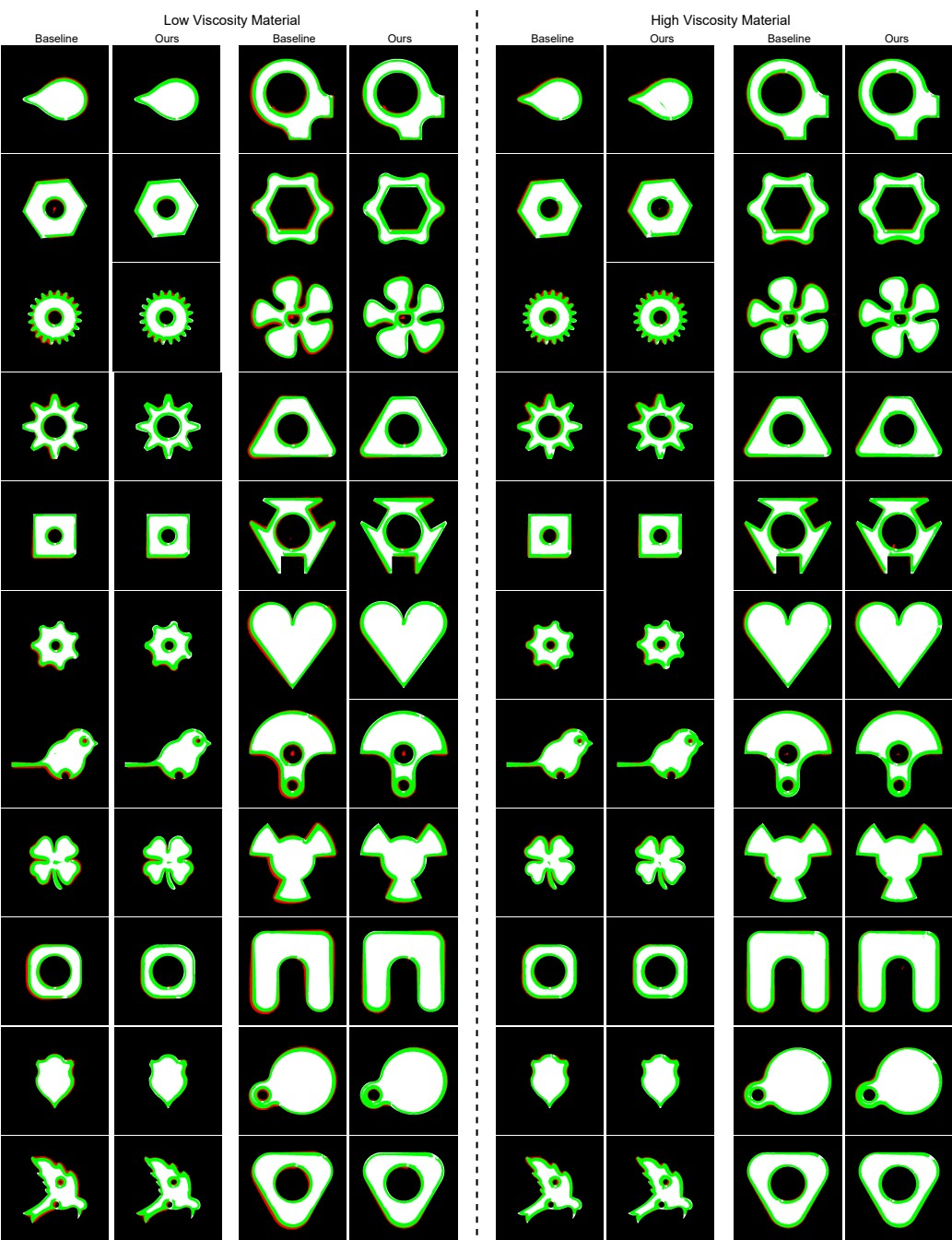

Figure 28: Policy evaluation on physical hardware.

