# OpenReview forum: "Closed-Loop Control of Additive Manufacturing via Reinforcement Learning"
_ICLR.cc/2022/Conference — ICLR 2022 Submitted_

### Official Review · Reviewer_WUp8 · 2021-10-22

**Correctness:** 3
**Technical Novelty And Significance:** 3
**Empirical Novelty And Significance:** 2
**Recommendation:** 5
**Confidence:** 3

**Main Review:**

Overall, this is a well-executed and evaluated system paper. I have learned a lot about the challenges of additive manufacturing from it. In addition, I found it very interesting that for such a complex system, end-to-end policies trained in simulation with off-the-shelf RL could work very well.
Along those lines, the main strengths of the paper are:
1. Very clear description of the proposed system, both in terms of hardware and software. The paper is well written and clearly presents the challenges of additive manufacturing.
2. An extensive evaluation of the sensorimotor controller both in simulation and in the real world.
3. Numerous ablation studies to support the design decision.

That being said, I think that the paper's main weakness is its very fine-grained specialization to the task of additive manufacturing. What is now completely missing is a discussion on how the key technical components ( e.g., learning with RL from a simplified model, selecting the action space, etc.)  could benefit other tasks in the realm of robot manipulation. A (non-exhaustive) set of possible questions could be:
1. Why is domain randomization not required to achieve sim-to-real transfer?
2. What parts of the simulated model need to match the real system closely, and what can be inaccurate?
3. What are the fundamental ingredients for transfer?
It would be extremely interesting to see these ideas generalized to other robots and tasks. In addition, some prior works observed that simple models are enough for training sensorimotor policies to control real robots (see for example [1] or [2]), so it would be nice to draw some connections with them as well.

A few more detailed feedbacks:

1. The related work section clearly shows that a lot of work has been done to improve additive manufacturing using traditional controller structures (like MPC) or end-to-end controllers. Why not include them in the experimental results? Having a quantitative comparison, at the very least in simulation, would strengthen the contribution.
2. The numerous ablation studies never show a comparison with a baseline. I think including baselines would facilitate understanding the failure cases of the ablated version of the approach.
3. The qualitative results are sometimes not very informative. For example, in Fig. 15, I have trouble understanding what works best. Maybe a better description focusing the attention of the reader in some specific parts could help.
4. (Minor) It would be nice to clarify why the approaches have higher failure rates for low viscosity, particularly the baseline controller.



[1] Deep Drone Acrobatics, Kaufmann, et al.
[2] Learning Dexterous In-Hand Manipulation, Andrychowicz et al.








**Summary Of The Paper:**

This paper proposes an approach to learn a closed-loop controller for additive manufacturing using reinforcement learning. The sensorimotor policy is trained exclusively in simulation and evaluated on a physical system in the real world without any fine-tuning with real-world data. To account for the difficulty to model the complex material deposition process, the approach builds a simplified simulator that can capture the process only qualitatively. Interestingly, even if only trained on the simple model, the approach generalizes to a real-world system.
The training methodology is described and evaluated in detail and compared against a baseline non-adaptive controller.

**Summary Of The Review:**

Overall, I found this to be interesting in both its technical execution and presentation. However, at the moment, it reads more as a technical report than a scientific paper. Learning how to extend the proposed technical innovation to other systems or tasks could strengthen the contribution, in my opinion.

---

> ### Author Response · Authors · 2021-11-17
> **Response to Reviewer WUp8 (1/2)**
>
> We thank the reviewers for their feedback. Based on the comments we suggest these improvements to paper’s exposition:
> * Restructure Section 4 and appendices into Appendix Methods to concentrate technical details in one place.
> * A new Section 4 that highlights key contributions to sim-to-real transfer in the context of related work, see the response to Reviewer WUp8.
> * Expand the discussion in the result section and include all clarifications.
>
> **How do the key technical components translate to other robot manipulation tasks?**
>
> An important insight we gained during this work is that  an engineered observation space coupled with learned features can significantly help with sim-to-real transfer. This insight suggests that to facilitate the learning process, it is worthwhile to invest in the design of observation spaces. A careful design of the observation space can improve the sim-to-real transfer, make the hardware design more flexible by enabling the use of a range of sensors that compute similar observations, and remove the need to hand-craft the features.
>
> We demonstrated that the policy learned in simulation could translate well to a physical task by employing a higher-level control scheme. This idea could also be generalized to other robotic tasks, for example, by applying a hierarchical divide and conquer approach to the action space. The control policies could output only high-level actions such as desired locations for robots actuators or deviations from a baseline behavior. Low-level controllers could then execute these higher-level actions. Such a control hierarchy can facilitate training by decoupling the higher-level goals from low-level inputs and transferring existing control policies to new devices through specialized low-level controllers.
>
> Our approximative transition function shows that it is not necessary to reproduce the physical world in simulation perfectly. A qualitative approximation is sufficient as long as we learn behavior patterns that translate to real-world experiences. This is an important observation for any task where we manipulate objects and elastic or frictional forces dominate the behavior. Relying on computationally more affordable simulations allows for applying existing learning algorithms to a broader range of problems where precise numerical modeling has prohibitive computational complexity. Moreover, by leveraging a numerical model it is possible to utilize privileged information that would be challenging if not impossible to collect in the real world.
>
> **What are the fundamental ingredients for successful transfer?**
>
> The three key ideas for a successful transfer are our abstractions on observation and action spaces coupled with randomization in the transition function.
>
> To facilitate the transfer of learning between simulation and a physical device, we rely on an abstraction of the observation space. Similar to [1], we do not use direct appearance values. Instead, in our case, we extract a heightmap. This allows our system to generalize to many different sensors such as cameras, depth sensors, or laser profilometers. However, unlike [1], we do not extract the feature vectors manually. Instead, similarly to [2], we learn the features directly from the heightmap. The critical difference with [2] is that we do not randomize the observation domain. Additional randomization is not necessary in our case thanks to the controlled observation conditions that facilitate observations.
>
> The reason why our abstracted action space facilitates sim-to-real transfer is similar in spirit to [2]. [2] suggest not using direct sensory inputs from the mechanical hand due to their noisiness and lack of generalization across environments. Instead, they use image data to track the robotic hand. Similarly, but instead in action space, we do not control the printer by directly inputting the typically noisy and hardware-specific voltages that actuate the motors of the apparatus. Instead, we control the printer by setting the desired velocity and offset and letting the apparatus match them to the best of its capabilities. This translation layer allows us to utilize the controller on a broader range of devices without per-device training.
>
> Lastly, to enable learning transfer, we apply randomization in the transition function. This is similar to [2], but instead of covering a large array of options, we specialize the randomization. Inspired by [3], we designed a data-driven LPC filter that matches the statistical distribution of variations observed during a typical printing process. This noise enables our control policies to adapt to changing environments and, to some extent, to changes in viscosity.
>
> **Improving Figure 15**
>
> We will improve the visualization of the physical results by scaling up the images and highlighting the deposition errors. We will support the visualization with a histogram comparing over/under-deposition of our method with the baseline.

---

> > ### Author Response · Authors · 2021-11-17
> > **Response to Reviewer WUp8 (2/2)**
> >
> > **Which parts of simulation can we approximate?**
> >
> > We have shown that the entire deposition process can be approximated. In our setting, we do not have the exact model for the deposited materials, the printing apparatus, or the contact interactions between the material, the printing nozzle, and the printing bed. Nevertheless, the policies learned in simulation translated well to the physical hardware.
> >
> > **Comparing with MPC**
> >
> > We purposefully did not compare with an MPC controller. The reason is that MPC relies on the quality of the predictions to formulate an action plan. This has two caveats. On the one hand, even our approximated model is too slow to be deployed on the physical hardware. On the other hand, by relying on an approximate model, the predictions will quickly deviate from actual material behavior.
> >
> > **Why is it harder to deposit lower viscosity materials?**
> >
> > Depositing lower viscosity materials is more challenging due to their behavior post-deposition. A material with high viscosity stabilizes within a short-horizon. In contrast, a low viscosity material will flow over long time horizons and interact with other deposited material, making behavior predictions significantly more challenging.
> >
> > [3] Chebotar et al., Closing the Sim-to-Real Loop: Adapting Simulation Randomization with Real World Experience, ICRA 2019

---

> > > ### Comment · Reviewer_WUp8 · 2021-11-22
> > > **Thanks for your answers**
> > >
> > > Thanks for clarifying some of my doubts in the manuscript. As already mentioned in my original review, the main problems remain that there are some strong claims (randomization not needed, the simulator is only qualitative, no need for a model of deposited materials and apparatus, etc.) that are not supported up by experiments. A system paper strongly depends on experiments, and therefore I will keep my score for now.

---

### Official Review · Reviewer_3pgj · 2021-10-30

**Correctness:** 4
**Technical Novelty And Significance:** 2
**Empirical Novelty And Significance:** 2
**Recommendation:** 5
**Confidence:** 3

**Main Review:**

Overall, this paper was pretty convincing in showing that the proposed approach is effective. My main concerns are (1) the aspect of the paper most directly relevant to ICLR, policy learning, used an existing algorithm and (2) the explanation of the approach was not entirely easy for me to follow.

I will readily admit that my lack of familiarity with additive manufacturing might be driving some of my confusion, but there were several parts in the modeling section of this paper that I found difficult to follow. In particular I found section 4.3 to be confusing and I think a few parts could be clarified there:
- Several times in this paper it says that the deposition process only needs to be captured "qualitatively". What does this mean exactly? What is this in contrast to? I think this needs to be explained in more detail.
- The discussion about time- vs. distance-based discretization was also confusing to me. I can understand some of the advantages of using distance-based discretization, but I find it hard to follow the part about gradient information vanishing and its relationship to Fig. 4. Can this be clarified?
- Fig. 5 and the associated discussion were also not clear to me.

**Summary Of The Paper:**

This paper presents a method for using RL for adjustment of process parameters in additive manufacturing. This allows for closed-loop control that outperforms the state-of-the-art in terms of printing quality. Multiple experiments showed this approach outperforming baselines, and it was also shown that the approach could be applied directly on physical hardware.

**Summary Of The Review:**

Given that this paper is application-focused and light on algorithmic contributions, I think there is should be a higher burden for it to be clear and readable. At the moment, I think the discussion about modeling is not clear enough. For this reason, I will recommend rejection, but will remain open-minded to the author rebuttal and the opinion of other reviewers.

---

> ### Author Response · Authors · 2021-11-17
> **Response to Reviewer 3pgj (1/1)**
>
> We thank the reviewers for their feedback. Based on the comments we suggest these improvements to paper’s exposition:
> * Restructure Section 4 and appendices into Appendix Methods to concentrate technical details in one place.
> * A new Section 4 that highlights key contributions to sim-to-real transfer in the context of related work, see the response to Reviewer WUp8.
> * Expand the discussion in the result section and include all clarifications.
>
> **Definition of qualitative observation and contrast with quantitative data**
>
> A qualitative simulation refers to a numerical model that is evaluated only subjectively. Typically, qualitative assessments are collected with user surveys. However, in our setting, a qualitative approximation would be considered good if it enables a control policy to navigate a real environment based on experience gathered in simulation.
>
> In contrast, a quantitative simulation can be evaluated objectively by comparing the simulated results with real-world ground truth. Unfortunately, utilizing quantitative numerical models is challenging as the numerous physical phenomena involved in additive manufacturing make them prohibitively expensive for learning.
>
> **Clarification why time-based discretization does not produce the desired gradient information**
>
> When employing a time-based discretization, the amount of deposited material is directly proportional to velocity. As we move slower, less material will be deposited within the same unit of time. Therefore, the immediate rewards for going slower are also lower. Since our control policies rely on these immediate rewards, the learning process favors larger immediate rewards and thus larger velocities. It is unclear if it can ever recover as this would require the final reward to backpropagate to the initial decisions. We have demonstrated that this is a challenging task in our ablation study on reward function.
>
> **Clarification on relationship between path direction and size of observation space**
>
> Imagine we trace the material only in the clockwise direction (Figure 5 bottom). When tracing the outer boundary we will observe that the material should be deposited to the left of the nozzle (Figure 5 bottom left).  In contrast, when we trace a hole inside the model, the material will be deposited on the right (Figure 5 bottom right). We can note that the set of outside outlines has the same elements as the set of holes since they can always be rescaled and swapped. As a result, the observation space contains the same samples just mirrored along the X-axis when following a single direction. If we did not treat this case, the control policy would need to infer the mirroring through the sheer amount of data. Instead, we remove this disambiguation by simply changing the holes' printing directions, which effectively mirrors the observation space (Figure 5 top right).

---

> > ### Comment · Reviewer_3pgj · 2021-11-29
> > **Thank you for the clarifications**
> >
> > Thank you to the authors for the clarifications, and for the restructuring/changes to the paper in response to reviewer comments.
> >
> > However, I will agree with the other reviewers and leave my score unchanged. As has been stated previously, this is an interesting paper and I encourage the authors to continue this line of work, but given the lack of algorithmic contributions there is a higher standard for clarity and experiments that should be met.

---

### Official Review · Reviewer_wpL1 · 2021-10-31

**Correctness:** 2
**Technical Novelty And Significance:** 2
**Empirical Novelty And Significance:** 3
**Recommendation:** 5
**Confidence:** 4

**Main Review:**

# Listing pros and cons
## Things I like about this paper:
- **an interesting application**: I think this paper addresses a known problem in the unusual but yet interesting application of 3D printing. Cheaper 3D printers are often imprecise which limits their usability in practice. Finding ways to make them more robust and capable can unlock a lot of useful applications which is great. Solving this with sim-to-real is also an exciting approach.
- **a thorough evaluation**: Studying the ability of the solution to work under different circumstances and providing a comprehensive ablation study is great. I commend the authors for the thorough evaluation of the proposed approach.
- **exciting architectural choices**: Using camera feed for closing the loop seems like a great choice, defining both the state space and the reward for the RL agent using the camera feed makes it appealing and potentially applicable in practice too.

## Things that can be improved:
- **ambiguous prose**: I found the paper generally hard to read, each section heavily depends on the appendix and a lot of the gluing details were missing. Furthermore, there are a lot of ambiguities in the description too. More details can be found below. A direct way to fix this limitation is to provide additional clarifications.
- **limiting assumptions**: The assumptions made in this work seem limiting at this point. The work evaluates on laying out only a single layer of a print assuming that there is a drastic difference in the pixel values between the printing bed and the depositing material but does not show the ability of the printer to complete an entire object's print. Currently, I doubt this is possible but yet, the paper claims to propose a fully functional self-correcting printer. If this is true, then perhaps toning down the claimed contribution and acknowledging the work's limitations can improve the clarity. Ideally, a working vision-based solution that is not limited to a single printed layer would be best. Perhaps this can be achieved by combining visual signal with the in-situ features mentioned in the paper and employed by prior work.
- **questionable improvements on the physical task**: Although the paper reports on the average offset improvement of the proposed solution with respect to a baseline controller, the provided photos of the results from the physical experiment do not seem different to the baseline controller. The lack of adaptation of the solution and its direct applicability from simulation to the physical world also brings up questions regarding the usability on the full 3D printing task. More details and clearer illustration of the provided results can help alleviate this issue.

# Claim and contribution
The overall paper claim seems to be a little too broad. The paper states that it introduces a first of its kind self-correcting printer but this is not true. Self-correcting printers is not that uncommon nowadays and works such as [1,2,3,4,5] seem to propose closed-loop control solutions too. In fact, [1] proposes a solution that actively controls the travel speed of printing by using an infra-red camera input, yet parallels were not drawn with any of these works. There also seem to be existing products that do similar things too - https://www.sciaky.com/additive-manufacturing/iriss-closed-loop-control. Perhaps, this can be further clarified

# Methodology
This work proposes to use PPO trained in simulation as an adaptive policy to a baseline controller for 3D printing. There are however a number of ambiguous statements that I detail below. Providing more details and clarifying these will make the prose and contribution much clearer.
## Observations
more details of what a heightmap is and how was the used segmentation obtained would be beneficial. Is the same RL policy used for both infilf and outline used. It seems like the images are assumed to be black and white and contain only 0s and 1s as values. Is this true, this seems limiting especially for applying multiple layers.
## Transition function
**using a model-free aproach**: Overall, I think it's great that there is so much information provided for the design and development of a reasonable transition function. However, given the amount of effort put to design a reasonable transition function, it is unclear why it is not being used in the learning process but instead the paper opts in for a model-free approach. Those approaches are generally useful in cases where the design and implementation of a reasonable transition function is impractical.

**modelling the deposition imperfections**: This paragraph is not very clear. I could not understand what is the purpose of modelling these imperfections reported in Figure 6 if they are already coming from a simulated and therefore controlled environment. The use of an LPC filter seems unnecessary but yet confusing. In addition, it is unclear if this information is provided to the agent or it's used for the simulation only.
## Reward function
The paper makes use of two types of rewarding- positive reward for depositing material inside the desired slice and a punishment term for depositing anywhere else. However, the work does not discuss how exactly are the rewarding and punishment terms combined together. The paper seems to make use of two different sets of rewards too: an infill and outline reward. However, it is currently unclear if the choice of using the infill or outline reward something that is manually assigned and whether this means that the same policy is trained using two different types of rewards? A discussion why this is ok/not ok would be beneficial. Furthermore, there is no additional information of what are the weights assigned on the outline reward. Calling the term W weights implies that those weights are also learnt, however there are no details associated with the specifics of how W is obtained or what values it takes all together. Finally, showing the resulted final reward, e.g. as a function of the rewarding and punishment terms can also be helpful to improve clarity.
## The use of a curriculum
Appendix F mentions the use of a curriculum of objects to train the RL agent. However, there were no details provided of the actual curriculum. Did the paper start training using simpler objects and gradually tried harder ones, or was there a different approach undertaken? More details can further clarify the training process undertaken.

# Evaluation

The paper provides a very thorough evaluation which is great, however the details and presentation of the obtained results can be further improved. I also have additional questions related to the obtained results. To begin with, there is no formal definition of an average offset. The verbal description is ambiguous and given that this metric is not a conventional metric for evaluation in the learning community some additional details, including a formal definition of the metric will be useful. Right now it is not clear if this metric can detect negative performance, such as falsely obtained improvements - e.g. spilling deposited material towards the infill part of the object as opposed to outside the outlines. In other words, it is not clear how the authors measure negative performance.

Figure 8 is the first time patterns are introduced in this work. However, it is not clear at all what the green lines show and why the left three plots are high gain and the right three plots are low gain. Details, including a more descriptive title will help understand this better. Broadly, the provided visual illustrations can be great to put things in perspective but are difficult source of measuring achieved performance when used on their own, especially for the provide ablations. Perhaps, reporting on the achieved average performance improvement in addition to the visual illustration can be useful. Right now, it is not very clear how much exactly 30microns of improvement is, for example, or how bad it is to get a little bit of spillage, as for example in Figure 12, last plot.

The description of the results obtained in Figure 12 are also ambiguous, what is it meant with 'On the other hand, modifying the printing path is slower and alone cannot handle sharp changes in material pressure'. How can this be seen from Figure 12, more details on this observation can be useful.

Section 5.1 compares against a refined solution that uses Bayesian optimistion as a preprocessing step to the execution, but it is unclear why this is done, is this a standard procedure in the field, then additional referencing can be helpful. The paper states that 'We can observe that the two control schemes require drastically different velocities' but it is not clear where this can be seen and how exactly.

Pre-test and post-test were not introduced anywhere in this work, it is not clear what is meant by those terms. The means and standard deviations are also confusing, what do these represent? The paper uses a holm-bonferroni test but does not provide additional details to what the total number of pairs is and what were their p-values. This is another non-standard metric for evaluation, additional details as to why this decision was made and what exactly it is supposed to measure would be helpful. What was the reason not to opt for more conventional and simpler metrics, such as the total average offset improvement, for example?

The varying pressure experiment does not provide sufficient details of how this pressure was varied and why this is a useful experiment. Right now it is not clear if this experiment is representative of the physical-world variations that occur in 3D printing. Perhaps evaluating this on hardware can be more informative, is there a reason to not do this?

Finally, the performance on the physical task shows some good results in terms of average offset improvement but those are not reflected in Figure 15. I find it hard to see why the proposed method is better than the baseline in this figure. However, it seems like this figure is the most important one from the whole paper as it showcases the applicability of the solution to physical tasks. Perhaps, highlighting and maybe further zooming in, as well as reporting the individual performance improvements as part of the figure can improve clarity.

# Additional details on the limiting assumptions
This work evaluates the proposed approach on a single layer of a 3D printed model. However, it does not disclose what would happen if the whole object's printout was to be done. Even though, a baseline solution is not as accurate on a single layer of a print, it will keep doing the same thing and arguably do a good job at printing the entire object, e.g. when applying more layers but I am curious to learn what will happen with the rl agent as right now it is not clear if it will succeed.

# Minor
- The choice of using scene units instead of cm seems a bit unconventional. Instead, relying on universally used metrics will make the work more readable.
- Thingi10K is misspelled to Thingy10K.
- The appendix should start on a new page and not directly after the references.
- The paper states that the policy can adapt to different viscosities but it also seems to assume individual learning procedures for the different viscosities. Therefore, a single learnt model cannot be adapted to different viscosities, instead the solution can be trained for different viscosities seems like a more accurate description of what it does.
- Fig. 4 description says left/right but should say first/second row
- Not good practice to describe a result in the main body of the paper but to keep the figure illustrating it in appendix.

# References
[1] Farshidianfar, Mohammad H. et al.  "Closed-loop control of microstructure and mechanical properties in additive manufacturing by directed energy deposition." _Materials Science and Engineering: A_ 803 (2021): 140483.

[2] Cerón Viveros et al. "Development of a closed-loop control system for the movements of the extruder and platform of a FDM 3D printing system." In _NIP & Digital Fabrication Conference_, vol. 2018, no. 1, pp. 176-181. Society for Imaging Science and Technology, 2018.

[3] Chen et al. "Data-Driven Adaptive Control for Laser-Based Additive Manufacturing with Automatic Controller Tuning." _Applied Sciences_ 10, no. 22 (2020): 7967.

[4] Freeman et al. "Beat the machine (learning): metal additive manufacturing and closed loop control." _Physics Education_ 55, no. 5 (2020): 055012.

[5] Razaviarab et al. "Smart additive manufacturing empowered by a closed-loop machine learning algorithm." In _Nano-, Bio-, Info-Tech Sensors and 3D Systems III_, vol. 10969, p. 109690H. International Society for Optics and Photonics, 2019.

**Summary Of The Paper:**

This paper demonstrates the feasibility of obtaining a closed-loop control policy using reinforcement learning for additive manufacturing also known as 3D printing. The paper proposes a sim-to-real approach that relies on a novel simulated environment that allows for synthesising successful off-the-shelve RL policies capable of improving upon existing state-of-the-art print deposition controllers. An underlying assumption of this work is to rely on qualitative information only assuming that the difference in colour between the background and the applied deposition is sufficiently large. The work is evaluated in both simulation and sim-to-real and shows that the proposed method works well under different circumstances in simulation such as using static or dynamic depositing pressure variations and depositing material with different viscosity. In addition, the paper provides a comprehensive ablation study over the choice of observation and action spaces as well as the choice of reward. Finally, this work demonstrates the applicability of the approach on a sim-to-real task with no additional fine-tuning showcasing its ability to achieve a higher offset improvement than a baseline controller. I think that this work proposes a reasonable solution to an interesting problem that can potentially impact the 3D printing industry. Although there was not necessarily a novel contribution from a learning perspective, I think this work proposes a novel solution to a curious application and is therefore worthy of consideration. However, I have additional questions and concerns that prevent me from recommending this work for acceptance yet. I detail those below.

**Summary Of The Review:**

In summary, I cannot recommend this paper for acceptance at its current state. Although the work introduces a potentially exciting solution to an interesting problem it has a number of ambiguous statements and lacks clarity on the otherwise extensive evaluation. In addition, some of the used assumptions in this work may prevent if from being actually applicable to full object 3D printing. Upon clearly and convincingly rebutting my review, I will be inclined to reconsider my recommendation.

---

> ### Author Response · Authors · 2021-11-17
> **Response to Reviewer wpL1 (1/3)**
>
> We thank the reviewers for their feedback. Based on the comments we suggest these improvements to paper’s exposition:
> * Restructure Section 4 and appendices into Appendix Methods to concentrate technical details in one place.
> * A new Section 4 that highlights key contributions to sim-to-real transfer in the context of related work, see the response to Reviewer WUp8.
> * Expand the discussion in the result section and include all clarifications.
>
> **System limitations and generalization to full 3D printing**
>
> We tested our system with single-layer printing, an essential task in many subfields of additive manufacturing. The single-layer deposition is used in bioprinting to deposit low viscosity hydrogels, in manufacturing of fiber optics and graphite sheets, or in assembly lines to deposit glue traces connecting individual parts. The main bottleneck preventing us from multi-layer printing is our acquisition module, where we extract the heightmap using the transparency of the deposited material. Such a system works reliably only up to a couple of layers. We opted for this system because it was more straightforward to implement.
>
> Given a different acquisition system capable of scanning the surface of higher layers, we believe our system would generalize well to multi-layer printing. Figure 10 presents the difference in filling an area with the baseline method and our controller. Our controller achieves lower over-deposition and creates an almost flat surface with no bulging. These improvements are significant in a full 3D print consisting of hundreds of layers. Since we did not perform multi-layer experiments, we will clarify in the paper that we focus on single-layer deposition.
>
> **Comparison with other self-correcting printers**
>
> We can categorize closed-loop printers depending on the stage of self-correcting behavior: hardware, process parameters, or physical printer output.
>
> The first category contains controllers that ensure the individual hardware components deliver consistent performance. An example is the stepper motor control in [2] or the pressure control in our printer. While this kind of control enhances the deposition consistency, it is not possible to correct for dynamic material behavior that depends on multiple process parameters.
>
> The second category improves material output consistency by tweaking one or more process parameters. Examples include tuning the velocity to maintain desired temperature profiles [1,3] or fixed filament width [Kim et al. Dynamic Control of a Fiber Manufacturing Process using Deep Reinforcement Learning, IEEE/ASME Transactions on Mechatronics 2021]. These systems typically observe a low-dimensional input signal (temperature, material width) and alter the process parameters that affect the deposition within a short horizon. The critical assumption is that controlling the material during deposition would lead to consistent printouts.
>
> In contrast, our work avoids such limiting assumptions. We seek to control the final output of a deposition process and guarantee high-quality printouts. To achieve our goal, it is not sufficient to control the material only during the deposition. It is also necessary to control how the material behaves after deposition as it interacts with other deposited materials over long horizons. These long-horizon interactions further complicate the problem as the final printer outputs are not perceived by an in-situ observer. To solve this problem, we use a fundamentally different formulation. We provide the learning algorithm with a rich observation space, an objective function quantifying good deposition, and let it learn the appropriate control policy. Such a formulation is significantly more challenging to learn as the features relevant for the process need to be learned together with the control policy. This makes simple control schemes commonly used by previous work insufficient in our scenario.
>
> **Improving Figure 15**
>
> We will improve the visualization of the physical results by scaling up the images and highlighting the deposition errors. We will support the visualization with a histogram comparing over/under-deposition of our method with the baseline.
>
> **Are there two different policies for outline and infill printing? How is the policy chosen at runtime?**
>
> There are two separate control policies - one for outline and one for infill printing. For a deposition system, it is a common practice to split these steps. This information is available in the slicer software, and we use it to pick the appropriate policy.
>
> **How is the reward combined with the punishment term?**
>
> Both reward and punishment terms are calculated on the same scale as the area of correctly-deposited and over-deposited material, respectively. To calculate the final reward, we sum them up to get a single number.

---

> > ### Author Response · Authors · 2021-11-17
> > **Response to Reviewer wpL1 (3/3)**
> >
> > **Clarification of discussion on ablation space and where can we see the effects on Figure 12?**
> >
> > Our action space consists of modifying the deposition velocity and adjusting an offset from the printing path. The first action is relatively fast and is constrained only by the motor acceleration scheme. In our apparatus, we can go from standing still to full speed in 6.6 milliseconds. In contrast, adjusting the deposition offset needs time for the printing head to move. Depending on the velocity, this can take anywhere from 0.13 to 1.3 seconds. As a result, both output actions have a different effect on the resulting print quality. In Figure 12 middle, we observe that modifying the velocity alone can not cope with significant material width variations. We can see it as the dots of over-deposited material around the boundary. Similarly, in Figure 12 left, we see that relying on offset alone is not sufficient. Due to the delay, the offset cannot react to fast changes. This is again visible as the more significant drops of over-deposited material. We will make these differences clearer in the revised version
> >
> > **Clarification on why Bayesian optimization was used to improve the baseline and result visualization.**
> >
> > To define our baseline, we followed industry standards to set the process parameters. However, it is possible that a different selection of process parameters would be more suitable to maximize our reward function. To check this hypothesis, we searched for optimal process parameters for two shapes. The results can be seen in Figure 26. We can observe that the process parameters obtained by Bayesian optimization are object-specific and do not transfer well across geometries. Moreover, our control policy achieves higher rewards thanks to its ability to tune process parameters on the fly.
> >
> > **What was the training curriculum?**
> >
> > We do not have a progressive curriculum. During each episode, a new object is selected at random from all the available candidates.
> >
> > **Clarification on the statistical analysis for improvement evaluation**
> >
> > A test refers to printing our evaluation dataset with a chosen policy and observing the final rewards. The pre-tests conditions are defined in the parenthesis, and they refer to our entire model with complete observation space, action space, and reward computation. The post-tests include modifications to one of the three variables mentioned above. The statistical test we conduct is a pairwise two-sided t-test that, in each case, shows that there is a significant difference between the samples. We do not report the exact P-values. Instead, we report an upper bound of all P-values from all three tests combined, i.e., the worst possible P-value. Since we conduct multiple comparisons, some tests could have erroneous outcomes, e.g., with a standard P-value threshold of 0.05 and 20 tests, it is expected that one would be statistically significant even though it is not. To counterbalance this effect, we employ the Holm-Bonferroni correction, which reduces the P-value threshold based on the number of observations.

---

> > > ### Comment · Reviewer_wpL1 · 2021-11-24
> > > **Thanks for the thorough replies!**
> > >
> > > Thanks for clarifying some of my doubts in the manuscript and for refining the prose. I appreciate the provided clarifications and effort put from the authors. I believe some parts of the paper have been drastically improved as a result.
> > >
> > > However, as already mentioned in my original review, the main problems remain. That is, the prose is still ambiguous and there are still some not supported strong claims. It seems like the refined version introduces additional ambiguous statements such as '...similar to OpenAI et al. observation but instead in action space...' and some strong claims not supported by experimentation, such as 'This idea could also be generalized to other robotic tasks, for example, by applying a hierarchical divide and conquer approach to the action space' or that the approach would work for more than 1 layer of deployed material - I do not see simplicity in implementation as a sufficient justification in a systems paper. The title and overall introduction also lays the foundations for a far broader solution, that works for more than a couple of layers of deployed material. Perhaps additional toning down of the prose and promised claims can be beneficial. Finally, I commend the authors for the new plots in Figure 10 but the reported results still seem marginally better than the baseline and I am not convinced that there are sufficient alternative baselines considered. For example, the rebuttal describes that the proposed transition function is a rough approximation and therefore not ideal to be used as part of an adaptive model-based solution but there are no comparisons to prove that.
> > >
> > > I share the views of Reviewer WUp8 that a systems paper strongly depends on experiments and thorough technical details, and therefore I will keep my score.

---

> > ### Author Response · Authors · 2021-11-17
> > **Response to Reviewer wpL1 (2/3)**
> >
> > **How is the heightmap defined? How does it differ between outline and infill?**
> >
> > The heightmap is a 2D image where each pixel stores the height of the deposited material. For each height map location, the height is measured as a distance from the building plate to the deposited material.
> >
> > We could use the heightmap as input to both policies. However, for outline printing we are interested only in our proximity to the boundary as such only a binary mask of the deposited material is sufficient.
> >
> > **Why opting for a model-free learning approach?**
> >
> > The proposed transition function is only a rough approximation of the deposition process. Learning the exact model dynamics will likely not result in adequate control as the proposed model is not suitable for predicting the behavior of a specific material. In contrast, relying on model-free learning and qualitative observations enables learning behavioral patterns that translate well to physical experiments.
> >
> > **Why is it necessary to model the deposition imperfections? What is the role of the LPC filter? Is the information provided to the agent or only to the simulation?**
> >
> > The materials in the simulated environment are perfectly homogenous. As a result, they consistently and repeatedly respond to a change in process parameters. Unfortunately, such an ideal assumption would not translate to the physical experiments where material and machine imperfection affects deposition quality. To introduce this element into training, we decided to use a data-driven pressure model in our transition function. The LPC filter generates material variations with similar statistical distribution as observed on the physical device. We based this approach on [Chebotar et al., Closing the Sim-to-Real Loop: Adapting Simulation Randomization with Real World Experience, ICRA 2019] that suggests that to facilitate sim-to-real transfer the randomization should have a similar distribution as in the physical world.
> >
> > **What are the weights assigned to the outline reward?**
> >
> > When printing the outline, we want to motivate the policy to deposit material as close to the model's edge as possible. We compute a map that modifies the reward for depositing material inside the target shape to achieve this. Since a matrix represents this map, we denote it W. W is computed by first calculating the distance transform of the target shape. The distance transform is a function that defines the distance towards the closest edge from each point within our target. Next, we threshold the distance function by a small value that limits rewarded deposition to a close neighborhood of the boundary.
> >
> > **Formal definition of average offset. How does it measure negative performance?**
> >
> > Given an image of the target slice T, printed canvas C, a weight mask W, and the length of the outline l, the average offset is computed as: ((1 - C)*T*W/l) + C*(1-T)/l which is jointly punishing the under- and over-deposition, respectively. The weight term limits the under-deposition calculation to a small distance from the boundary. As a result, the control policy is motivated to closely match the boundary but not punished for spilling material inside the part.
> >
> > **Figure8: what do the colors mean? Where should we focus to appreciate the differences? How much difference is significant?**
> >
> > The black and white color marks the non-part, part areas, respectively. The goal of each policy is to fill the part area as well as possible without causing over-deposition. We visualize this in two colors. Green marks material deposited inside the part and red marks material deposited outside of the part. The comparison between high and low gain structures shows the difference between depositing a smooth rounded part and a jaggy structure. We can observe that the sharp and thin corners cause local under and over deposition. In contrast, the smooth curvature regions lead to minimal over or under-deposition. We will make this clearer in a revised version of the figure.
> >
> > **Details on physical experiments and why we opted for a predefined pressure variation**
> >
> > The pressure was varied using an oscilloscope by setting a sinusoidal signal to the pressure control of our 3D printer. The amplitude of the sinusoid was 8-14 psi, and the period was 47 seconds. By utilizing a computer-controlled pressure variation, we can guarantee consistency across the trials. This allows us to quantify the improvement of our control policy over the baseline.
> >
> > **Sensitivity to different viscosities**
> >
> > The model can be trained to handle different viscosities is the correct formulation. However, while we did not explicitly evaluate this, an individual controller can adapt within a specific viscosity range. We will provide examples of how a control policy can cope with viscosity change in the revised version.

---

### Author Response · Authors · 2021-11-22
**Summary of revision**

Dear Reviewers,

We have prepared a preview of the changes promised in the rebuttal (highlighted as red text). Most importantly:
- We moved former Section 4 into Appendix A and created a new Section 4 highlighting our core contributions and the applicability of our findings to other control tasks.
- We updated the description of average offset to include the formal definition in Equation 1
- Updated former Figure 15 (now Figure 10) to better visualize the improvement of our control policy on the physical hardware when compared with the baseline. More specifically, we plot histograms showing the better behavior of our deposition process.
- Created a new Figure 27 in Appendix C that visualizes how our control policies behave on materials with varying viscosity.

---

### Decision · Program_Chairs · 2022-01-20

**Decision:**

Reject

**Comment:**

All reviewers agreed that the paper contains interesting experiments. However, as this paper is a systems paper without much algorithmic contributions, all reviewers felt that the paper felt short in terms of describing the results, has too many unsupported claims and it is unclear how the presented results transfer to slightly different domains. I therefore agree with the reviewers and recommend rejection of the paper.